# Research Progress of Ribosomal Proteins in Reproductive Development

**DOI:** 10.3390/ijms252313151

**Published:** 2024-12-07

**Authors:** Yuqi Hong, Qisheng Lin, Yuan Zhang, Jilong Liu, Zhanhong Zheng

**Affiliations:** College of Veterinary Medicine, South China Agricultural University, Guangzhou 510642, China; lhongyuqi@163.com (Y.H.); lqs@stu.scau.edu.cn (Q.L.); zhangyuan@scau.edu.cn (Y.Z.); jilongliu@scau.edu.cn (J.L.)

**Keywords:** ribosomal proteins, reproduction, stress, proliferation, autophagy

## Abstract

Ribosomal proteins constitute the principal components of ribosomes, and their functions span a wide spectrum. Recent investigations have unveiled their involvement in oocyte and embryo development, playing a pivotal role in reproductive development. Numerous pieces of evidence indicate that ribosomal proteins participate in the regulation of various cellular activities, including nucleolar stress, oxidative stress, cell proliferation and autophagy. Despite these findings, the precise mechanisms through which ribosomal proteins influence reproductive development via these cellular activities remain elusive. Therefore, elucidating the mechanisms of action is essential for a comprehensive understanding of the role and function of ribosomal proteins in reproductive development. This paper systematically reviews the progress in research on nucleolar stress, oxidative stress, cell proliferation and autophagy concerning ribosomal proteins during reproductive development. Furthermore, we explore the potential of ribosomal proteins as diagnostic markers for various diseases. Additionally, we propose the development of drugs and therapies targeting ribosomal proteins, underscoring the potential for novel medical interventions in the context of reproductive health.

## 1. Introduction

Infertility has emerged as a significant societal concern in recent years, with its increasing global prevalence attributed to a myriad of factors, including age at childbearing, occupational factors, substance abuse, environmental pollution, sexual infections and lifestyle choices. Notably, this condition is affecting individuals at increasingly younger ages [1]. In the United States, approximately 10% of women experiences infertility [2]. A population-based epidemiological survey encompassing eight provinces in China unveiled a concerning prevalence of infertility, with a staggering 15.5% prevalence rate observed among couples cohabiting for over a year, where wives were aged 20 to 49 [3]. In Henan Province specifically, the prevalence of infertility among women of childbearing age stands at 24.58%, with primary and secondary infertility rates documented at 6.54% and 18.04%, respectively [4].

Reproduction constitutes a complex process encompassing various stages, including sperm development, oocyte development, sperm–egg fusion, embryo implantation, decidualization, placenta formation and the maintenance of pregnancy [5,6]. It is noteworthy that each of these stages involves cell proliferation, differentiation and development, all of which entail significant protein engagement [7]. Consequently, ribosome biosynthesis increases within cells to meet this heightened demand during reproductive development [8]. Emerging research underscores the regulatory roles played by ribosomal proteins in ribosome biosynthesis, as well as their pivotal significance in oocyte development, sperm development and embryonic growth [9,10,11]. These findings underscore the critical importance of ribosomal proteins in the realm of reproductive development. In this review, we provided an extensive overview of relevant studies investigating the roles of ribosomal proteins in reproductive development, with a particular focus on recent advances in our understanding of nucleolar stress, oxidative stress, cell proliferation and autophagy of these proteins.

## 2. Classification and Function of Ribosomal Proteins

Ribosomal proteins (RPs) represent the fundamental constituents of ribosomes [12]. In bacterial ribosomes, the formation of small subunits involves 21 ribosomal proteins, while 33 ribosomal proteins contribute to the constitution of large subunits. Conversely, eukaryotic ribosomes comprise 33 ribosomal proteins in small subunits and 47 in large subunits [13]. Ribosomal proteins associated with the assembly of the 40 S subunit are designated as RPS, whereas those involved in the assembly of the 60S subunit are denoted as RPL [14].

Ribosomal proteins serve a myriad of functions [15]. Their essential roles encompass participation in ribosome subunit assembly and ribosome biogenesis [8]. Additionally, ribosomal proteins exhibit extra functions, such as involvement in cell cycle regulation, DNA repair, antimicrobial and antiviral activities [16,17,18]. For example, ribosomal proteins are involved in the p53 pathway induced by nucleolar stress; binding to Mouse double minute (MDM2), thereby mediating p53 stabilization and inducing nucleolar stress and cell cycle arrest [19]. In addition, RPLP0 binds to the RRM1 and RRM2 domains of NONO to enhance non-homologous end-ligation-mediated DNA double-strand break repair [20]. Some studies suggest that ribosomal proteins possess antibacterial properties. RPS23 and RPS15 have been demonstrated to exhibit antimicrobial peptide activity, capable of eliminating both Gram-negative and Gram-positive bacteria [16,21,22]. Furthermore, specific ribosomal proteins can interact with viral proteins to neutralize viruses and exert antiviral effects. For instance, RPS10 can interact with swine fever virus protein Npro, inhibiting swine fever virus replication [23]. However, some ribosomal proteins can directly bind to viral mRNA to promote viral proliferation. For example, the mRNA of vesicular stomatitis virus may directly bind to RPL40, or RPL40 may cause conformational changes in the 60S subunit to promote preferential translation of viral mRNA [24]. Therefore, ribosomal proteins may play different roles in different viral replication. It is noteworthy that the absence or mutation of ribosomal protein genes can result in defective ribosomal protein expression, potentially leading to ribosomal diseases. Approximately 50% of Diamond–Blackfan anemia (DBA) patients exhibit gene mutation in ribosomal proteins, with congenital loss-of-function mutations in RPS19, RPS10, RPL11 and RPL5 being the most prevalent [25].

Ribosomal proteins have emerged as promising diagnostic markers for various diseases. For instance, RPL5 and RPL10 have been identified as potential diagnostic markers for atypical teratoid/rhabdomyoid tumors [26]. Furthermore, a significant upregulation of RPL22L1 and RPS21 expression in prostate cancer tissues was observed, suggesting their potential utility as diagnostic markers for prostate cancer [27]. In the context of gastric cancer, RPL15, RPL6 and RPS13 exhibited upregulated expressions, while in colorectal cancer, RPS18, RPS23, RPL28 and RPL32 were found to be downregulated, implicating these ribosomal proteins as potential diagnostic markers for gastric and colorectal cancers, respectively [18]. Notably, RPS24 plays a pivotal role in human colon cancer, with its knockdown leading to significant inhibition of colon cancer cell proliferation, suggesting its potential application in colon cancer diagnosis [28]. Moreover, RPL23A has demonstrated promise as a prognostic marker for rectal cancer [29]. These findings underscore the diagnostic potential of ribosomal proteins in various malignancies.

## 3. Ribosomal Proteins and Reproductive Development

RPS3 is expressed highly in oocytes and preimplantation embryos. Knocking out RPS3 in fertilized eggs led to a developmental arrest, which prevented progression to the blastocyst stage. These findings prove the crucial role of RPS3 in early embryonic development [30]. Another investigation observed that in developing oocytes, the inactivation of one allele of RPS6 resulted in the generation of S6 heterozygous embryos. Notably, oogenesis and embryonic development proceeded normally until embryonic day 5.5. However, after this stage, a cell cycle arrest occurred during the M phase, accompanied by defective cell proliferation and increased apoptosis. The embryos finally died at the gastrula stage. These results suggest that RPS6 plays an indispensable role in oocyte development. Nevertheless, further research is needed to elucidate the mechanism by which RPS6 influences embryonic cell survival [31]. Similarly, mitochondrial ribosomal proteins (MRPs) including MRPL3, MRPL22, MRPL44, MRP18c and MRPS22 have been knocked down in mutant mouse embryos. It was found that the embryos failed to initiate gastrulation at day 7.5, leading to embryonic demise. Notably, these five mitochondrial ribosomal proteins exhibited abundant expression both before and after implantation in normal mouse embryos, suggesting their potential contribution to gastrulation [32]. Furthermore, the deficiency of RPL10a was found to result in embryonic developmental delay, abnormal embryonic morphology and increased apoptosis in zebrafish [33]. In another study by Sijia et al., interference with the expression of RPL11 and RPS2 led to a significant decrease in the oviposition rate, egg-hatching rate and fecundity of Phytoseiidae, suggesting the potential involvement of RPL11 and RPS2 in egg formation and the development of Phytoseiidae [34]. In higher plant cells, RPL14B was found to guide pollen tube growth during fertilization, a critical process for successful fertilization in higher plants. Therefore, RPL14B plays an essential role in reproduction [35].

In summary, ribosomal proteins emerge as pivotal contributors to various reproductive activities across diverse organisms (as shown in Table 1). However, the intricate molecular mechanisms underpinning their regulatory roles remain incompletely elucidated. Consequently, there is a compelling interest in deciphering the precise mechanisms through which ribosomal proteins exert their influence on reproductive development. This endeavor promises to enhance our comprehension of the multifaceted roles and functions of ribosomal proteins within the realm of reproductive development. Hence, this review delves into the extant literature on ribosomal proteins, offering a comprehensive overview of the progress made in understanding their involvement in reproductive development. Specifically, as shown in Figure 1, we have distilled their impact on three fundamental facets of this process: (1) cellular stress, (2) cell proliferation and (3) cell autophagy. By exploring these dimensions, we aim to provide the advancements in ribosomal proteins’ contributions to reproductive development.

### 3.1. Influence of Ribosomal Proteins on Stress

#### 3.1.1. Influence of Ribosomal Proteins on Nucleolar Stress

When cells are exposed to various stressors such as hypoxia, toxins and nutrient deprivation, they undergo a condition known as nucleolar stress, also referred to as ribosome stress [49]. During this state, the nucleolus experiences morphological alterations and functional deficiencies, ultimately culminating in cell cycle arrest, apoptosis or autophagy dysfunction through the activation of the p53 signaling pathway or other stress signals [50]. Donati and Havel et al. have provided evidence that when nucleolar stress occurs, RPL11 relocates from the nucleolus to the nucleoplasm, where it forms a complex with RPL5 and 5S rRNA. This complex binds to MDM2 in the nucleoplasm, preventing the ubiquitination and degradation of p53, thereby activating the p53 pathway [51,52]. Furthermore, RPS27a has been identified as a novel RPL11-binding protein. Knockdown of RPS27a reduces its interaction with RPL11 but promotes the binding of RPL11 to MDM2, resulting in p53 activation, cell cycle arrest and apoptosis [53]. Additionally, RPS25 interacts with MDM2, inhibiting its E3 ligase activity, leading to p53 stabilization and activation. Knockdown of RPS25 attenuates p53 activation following nucleolar stress [54]. Notably, deletion of 24 out of 80 human ribosomal proteins activates p53, leading to a 5- to 10-fold increase in p53 levels [55].

Some studies have highlighted the significance of ribosomal protein loss in activating the p53 pathway and its consequences on embryonic development and survival. Knockdown of RPL11 expression in zebrafish resulted in p53 pathway activation and aberrant brain development in RPL11-deficient embryos. This condition was characterized by widespread apoptosis and the upregulation of genes associated with the p53 pathway, ultimately leading to embryonic mortality [38]. Similarly, in RPS7-deficient zebrafish embryos, p53 activation was observed, accompanied by apoptosis and cell cycle arrest. Moreover, RPS7 deficiency severely impaired hematopoietic function in these embryos, resulting in developmental anomalies [39]. Furthermore, DNA damage in spermatogonia induced by carbon ion radiation could trigger nucleolar stress, down-regulating the expression of RPL27a. The reduction in RPL27a levels, in turn, activated p53, leading to spermatogonia apoptosis and ultimately causing male infertility [40].

Embryo implantation and decidualization are crucial processes in reproductive development. However, the current scientific literature lacks comprehensive reports on the regulatory role of ribosomal proteins in nucleolar stress in these events. It was revealed that nucleolar stress occurs during mouse embryo implantation induced by actinomycin D. Nucleolar stress was also detected in uterine cavity epithelial cells during the receptive state, leading to activation of the p53 pathway. Interestingly, in human endometrial epithelial cells, actinomycin D-induced nucleolar stress promoted embryonic adhesion, suggesting the presence of nucleolar stress during early pregnancy in both mice and humans, and its potential to induce embryo implantation [56]. Furthermore, the low-dose actinomycin D-induced nucleolar stress can facilitate the transition from a matrix to an epithelial state, thereby promoting decidualization [57]. Consequently, it remains an intriguing question whether ribosomal proteins can promote embryo implantation and decidualization through the activation of the p53 pathway. This hypothesis requires additional experimental data for validation and a more in-depth understanding.

#### 3.1.2. Influence of Ribosomal Proteins on Oxidative Stress

Oxidative stress arises from an imbalance characterized by elevated levels of reactive oxygen species (ROS) and a deficiency of antioxidants, leading to physiological irregularities within cells, ultimately resulting in cellular dysfunction and death. An increasing body of research has established that oxidative stress is not limited to diseases such as inflammation and cancer but also extends to reproductive disorders [58]. Persistent and potent oxidative stress has been reported to compromise the antioxidant function of the placenta, leading to damage to lipids, proteins and DNA within placental tissue. This, in turn, accelerates premature placental aging and insufficiency, adversely impacting fetal viability [59]. Moreover, it has been observed that infertile men exhibit higher levels of ROS in their semen compared to their fertile counterparts. Excessive ROS in semen can impair sperm activity, cause DNA damage and result in compromised sperm function, thereby contributing to male reproductive dysfunction. Consequently, oxidative stress is considered a pivotal factor in male infertility [60,61]. These findings show the profound impact of oxidative stress on the reproductive and developmental processes of the body, highlighting the significance of investigating the relationship between ribosomal proteins and oxidative stress.

It has been reported that exposure to low concentrations of hydrogen peroxide induces oxidative stress in HeLa cells, resulting in the carbonylation of RPL32 and RPL35 within these cells. The carbonylated forms of RPL32 and RPL35 undergo subsequent regulation by apoptosis and autophagy, likely as a mechanism to impede the proliferation of cells that have incurred damage due to oxidative stress [62]. Furthermore, exposure to low levels of hydrogen peroxide has been shown to significantly reduce the rate of blastocyst formation and impede embryonic development [63]. In a separate study, the downregulation of MRPL24 expression in *Caenorhabditis elegans* resulted in alterations in mitochondrial function, subsequently triggering an oxidative stress response mediated by SKN-1. This genetic intervention led to marked developmental delays, reduced size and diminished offspring production in the larvae [41]. Moreover, investigations have indicated that the knockdown of RPS9M disrupts mitochondrial ribosome biogenesis and increases ROS production, culminating in oxidative stress-induced embryonic abnormalities and defects. These findings underscore the pivotal role of RPS9M in Arabidopsis male gametogenesis and seed development [42].

The regulatory effect of ribosomal proteins on oxidative stress is not only reflected in the process of reproductive development but also plays a role in the occurrence of other diseases. A case in point is Diamond–Blackfan anemia, a condition characterized by oxidative stress and predominantly linked to pathogenic variants within ribosomal protein genes [64]. Kapralova et al. discovered that the absence of RPL5 and RPS19 in mouse erythroleukemia cells led to the downregulation of superoxide dismutase and catalase expression, concomitant with elevated ROS levels. This observation suggests that the expression of RPL5 and RPS19 is intricately associated with cellular antioxidant capacity, potentially rendering cells less resistant to excessive ROS levels upon their loss [65]. Furthermore, Prakash et al. ascertained that RPSB possesses antioxidative stress properties. Specifically, mycobacteria expressing RPSB exhibited enhanced resilience to oxidative stress induced by 2.5 mM hydrogen peroxide, 0.05% SDS and starvation. Additionally, these RPSB-expressing mycobacteria reduced levels of drug-induced ROS. This finding underscores the potential role of RPSB in conferring anti-stress capabilities to mycobacterial species [66].

### 3.2. Influence of Ribosomal Proteins on Proliferation

RPL5 has been reported to stimulate the proliferation of colon cells through the activation of the MAPK/ERK signaling pathway [67]. Similarly, RPL22L1 has the capability to activate the PI3K/Akt/mTOR pathway, promoting the proliferation of prostate cancer cells, offering a promising avenue for prostate cancer therapy [68]. Overexpression of RPS15A can activate the mTOR pathway and promote the proliferation of gastric cancer cells, thereby promoting the progression of gastric cancer [69]. Knockdown of RPS6 results in cell cycle arrest at the G0/G1 phase, ultimately restraining the proliferation of ovarian cancer cells [37]. Notably, following the knockdown of RPL39 in trophoblast cells, an increase in the number of cells in the G0/G1 phase and a decrease in the number of cells in the S phase occur, thereby implicating RPL39 in cell cycle regulation and the inhibition of trophoblast cell proliferation [44]. Additionally, RPL10A induces insulin receptor overexpression to activate insulin signaling, affecting cell proliferation [70]. Clinical evidence reveals the high expression of RPL34 in non-small-cell lung cancer (NSCLC) patient samples, and the overexpression of RPL34 has been shown to promote malignant proliferation in NSCLC cells [71]. Moreover, in mouse cochlea, the overexpression of RPS14 has been linked to enhanced Sertoli cell proliferation through the activation of the WNT signaling pathway [45]. Collectively, these findings underscore the multifaceted roles of ribosomal proteins in the regulation of cell proliferation across diverse signaling pathways. It is noteworthy that the WNT/β-catenin pathway plays an important role in various aspects of embryonic development, particularly during early embryonic development and placental development. Additionally, the WNT/β-catenin pathway governs the proliferation of embryonic stem cells and augments embryo implantation potential [72,73]. The WNT/β-catenin pathway has been recognized for its regulatory roles in physiological processes, such as cell proliferation, differentiation and invasion [74]. Dysregulation of the WNT/β-catenin pathway in the endometrium can result in embryo implantation failure and severe pathological changes in the endometrium, including conditions like endometrial cancer and endometriosis [75]. Hence, exploring the association between ribosomal proteins and the WNT/β-catenin pathway in the context of reproductive development is of paramount importance.

Specifically, the knockdown of RPL10 resulted in a reduction of TCF4/LEF1 protein levels, effectively inhibiting the WNT/β-catenin pathway [76]. In hepatocellular carcinoma, RPS15A was found to promote angiogenesis by enhancing WNT/β-catenin-induced FGF18 expression. Significantly, inhibiting RPS15A expression markedly impeded tumor growth and angiogenesis in tumors [77]. Moreover, RPS15A exhibits high expression in pancreatic cancer cell lines and facilitates the proliferation of pancreatic cancer cells through the WNT/β-catenin pathway [78]. Knockdown of RPS15A was also shown to inhibit the WNT/β-catenin signaling pathway in ovarian cancer cells [43]. Within the MYC family, c-MYC is intricately involved in ribosomal protein synthesis and is a direct target of β-catenin [14]. However, it is worth noting that RPL5, RPL11 and RPS14 can inhibit the transcriptional activity of c-MYC [79,80,81]. Furthermore, RPL5 and RPL11 exhibit synergistic effects in down-regulating c-MYC expression through a negative feedback regulation mechanism [82]. Additionally, the inactivation of APC has been shown to dysregulate the WNT/β-catenin signaling pathway, leading to overexpression of c-MYC, RPL11 and RPL5 [83]. In short, ribosomal proteins are strongly intertwined with the WNT/β-catenin signaling pathway. Nonetheless, our understanding of the molecular mechanisms through which ribosomal proteins engage with the WNT/β-catenin signaling pathway during reproductive development remains limited.

### 3.3. Influence of Ribosomal Proteins on Autophagy

Autophagy, a catabolic process triggered in response to various cellular stressors, serves as a protective mechanism against diverse cytotoxic insults [84]. In recent years, mounting evidence has illustrated the significance of autophagy dysregulation in an array of human diseases, encompassing neurodegenerative conditions, infectious ailments, autoimmune disorders, as well as various cancers [85]. Notably, it is imperative to recognize the pivotal role that autophagy plays in female reproduction [86]. Studies by Su et al. have emphasized the essential role of endometrial autophagy in early pregnancy and its association with early pregnancy endometriosis, which can result in implantation abnormalities in embryos [87]. In addition, the autophagy gene Atg16L1 is indispensable for the decidualization of the endometrium. In the absence of Atg16L1, the number of embryos capable of successful implantation was significantly reduced and mouse endometrial stromal cells failed to undergo proper decidualization. Furthermore, interference with Atg16L1 expression in human endometrial stromal cells downregulated the expression of decidualization markers, including RPL and IGFBP1 [88]. Consequently, the development of safe and effective natural products capable of regulating endometrial autophagy has been proposed as a novel treatment strategy for female infertility [89]. It is also important to note that autophagy plays a crucial role in male reproduction. ARMC3, for instance, is beneficial for autophagy activity during spermatogenesis. The deficiency of ARMC3 can impede ribosomal autophagy, resulting in an elevated level of ribosomal proteins and eventual sperm immobility, ultimately leading to complete male infertility [90].

Decreased mTOR activity, reduced RPS6 phosphorylation and increased autophagy were found in hypoxic human primary trophoblast cells. This suggests that autophagy may enhance placental function in vivo by eliminating organelles damaged due to hypoxia [36]. In the context of pregnancy complications, such as pre-eclampsia, characterized by impaired autophagy, treatment of autophagy-deficient trophoblasts with serum from pre-eclampsia patients led to elevated levels of phosphorylated RPS6KB, increased rapamycin complex 1 (mTORC1) activity and the accumulation of placental protein aggregates. These findings may contribute to adverse pregnancy outcomes [91]. Additionally, the expression of RPS7 in red blood cells of zebrafish embryos was linked to increased cell autophagy [92]. In placentas from intrauterine growth restriction pregnancies, decreased protein expression of RPL26 and RPS10 positively correlated with mTORC1 signaling [93]. Recently, it has been reported that RPL5 inhibits breast cancer cell growth by regulating ER stress and autophagy through E2F1 in breast cancer cells and tissues [46]. Moreover, the down-regulation of RPLP proteins, including RPLP0, RPLP1 and RPLP2, resulted in reduced proliferation, cell cycle arrest and induced autophagy in breast and ovarian cancer cells [47]. It was reported that interfering with RPS27L expression shortens the protein half-life of β-TrCP, leads to the accumulation of DEPTOR to inactivate mTORC1 and finally significantly induces autophagy in breast cancer cells and mouse fibroblasts [48]. In summary, decreased expression of ribosomal proteins leads to increased autophagy during reproductive development. Interestingly, the same regulatory effect of ribosomal protein expression on autophagy seems to exist during the study of other diseases. Similarly, Rps23-deficient HEK293 cells were susceptible to proteotoxic stress, with reduced cell viability and a three-fold decrease in apoptosis rate after treatment with the autophagy activator rapamycin [94]. Knockdown of RSL1D1 promoted autophagy, invasion and migration of colon cancer cells, while overexpression of RSL1D1 inhibited autophagy in colon cancer cells [95]. Collectively, these studies highlight the key role of normal ribosomal protein expression in maintaining autophagic homeostasis.

## 4. Conclusions

Statistical data reveal that the monthly success rate of human pregnancy stands at approximately 30%, with a significant portion of pregnancy failures attributed to unsuccessful embryo implantation [96]. Ribosomal proteins are involved in the regulation of a variety of cell activities, and their roles in reproductive development have also been reported. However, there are few reports on specific reproductive activities such as embryo implantation and decidualization. Undoubtedly, embryo implantation and decidualization are the key events in reproductive development. Therefore, it is of great interest to study the role of ribosomal proteins in the process of embryo implantation and decidualization and further reveal the specific molecular mechanism of ribosomal proteins.

Cell stress, cell proliferation and autophagy are important activities for the normal operation of cells, which are closely related to the reproductive and developmental process of the body. Many studies have suggested that ribosomal proteins are involved in cell stress, cell proliferation and autophagy, thereby exerting influence on reproductive development. In terms of cell stress, existing studies have predominantly concentrated on exploring the association between ribosomal protein loss and defects in embryonic development and embryo mortality; there remains a dearth of investigations into their involvement in other reproductive events, particularly embryo implantation and decidualization. Therefore, an endeavor can be undertaken to elucidate whether ribosomal proteins contribute to embryo implantation and decidualization by instigating nucleolar stress. Concerning cell proliferation, ribosomal proteins are known to regulate this process through diverse signaling pathways. A growing body of evidence underscores the close relationship between ribosomal proteins and the WNT/β-catenin pathway. Therefore, future research avenues could go into whether ribosomal proteins are downstream targets of this pathway. In the realm of autophagy, the loss of ribosomal proteins has been associated with autophagy activation, primarily within the placental context. However, the variety of ribosomal proteins involved in this process is limited. Future research endeavors may unearth additional ribosomal proteins implicated in placental autophagy, thereby shedding further light on the molecular mechanisms underlying the relationship between ribosomal proteins and autophagy. In addition, the relationship between ribosomal proteins and autophagy during embryo implantation and decidualization has not been reported, so this is also worthy of further exploration.

In the future, it is of great interest to investigate the specific regulatory mechanisms of ribosomal proteins on cellular stress, cell proliferation and autophagy during reproductive development. It has been reported that the expression of MRPL15 is significantly increased in ovarian cancer patients and is associated with poor prognosis of patients. MRPL15 may play a role in ovarian cancer through cell cycle or DNA repair pathways. Therefore, MRPL15 can be used as a new prognostic biomarker and therapeutic target for epithelial ovarian cancer [97]. Furthermore, compared with normal tissues, RPLP0, RPLP1and RPLP2 are significantly up-regulated in gynecologic tumors such as ovarian and endometrial cancers, suggesting that these proteins can be used as specific prognostic markers and related therapeutic targets for gynecological tumors [98]. Therefore, the development of medicine and therapeutic methods targeting ribosomal proteins may be an effective strategy for the treatment of reproductive and developmental diseases caused by cell stress, cell proliferation, autophagy and other disorders. As shown in Figure 2, we mapped some ribosomal proteins that could serve as biomarkers in reproductive diseases. In summary, an in-depth exploration of the specific functions of ribosomal proteins in reproductive development holds significant promise for treating infertility-related conditions and enhancing clinical pregnancy rates.

## Figures and Tables

**Figure 1 ijms-25-13151-f001:**
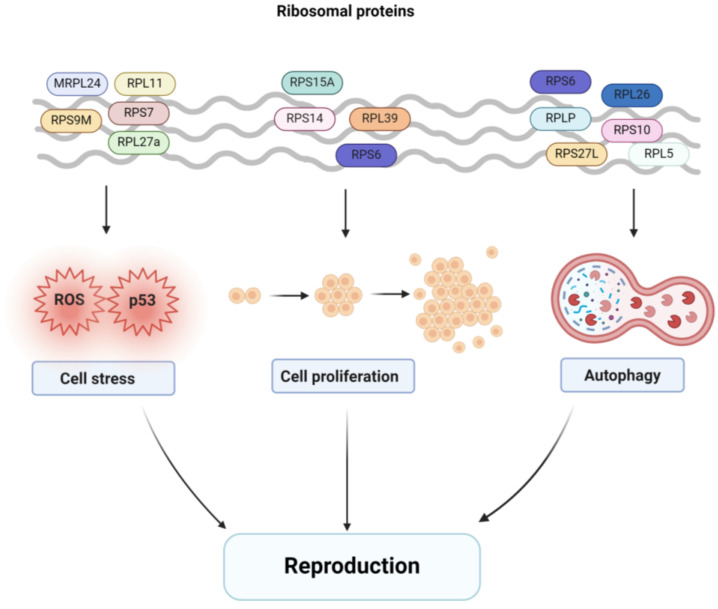
Ribosomal proteins affect reproductive development processes through cell stress, cell proliferation and autophagy.

**Figure 2 ijms-25-13151-f002:**
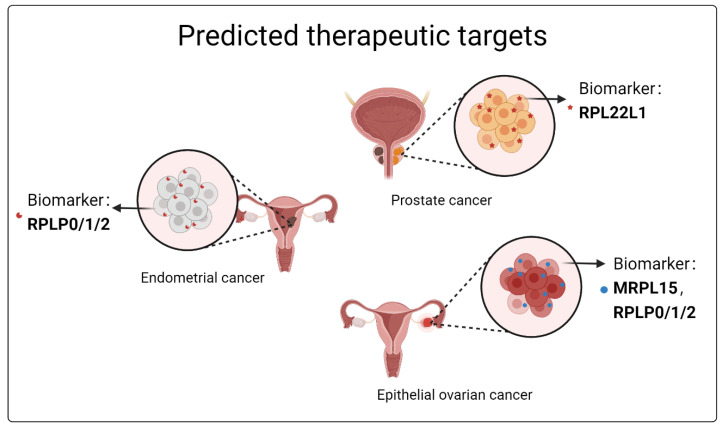
Predicted therapeutic targets of prostate cancer, endometrial cancer and epithelial ovarian cancer.

**Table 1 ijms-25-13151-t001:** Some ribosomal proteins involved in reproductive and developmental events.

Ribosomal Proteins	Events of Reproductive Development	References
RPS3	Early embryonic development	[30]
RPS6	Oocyte development, ovarian cancer, placental function	[31,36,37]
MRPL3, MRPL22, MRPL44, MRP18c, MRPS22	Gastrulation	[32]
RPL10a	Embryonic developmental delay	[33]
RPL11	Egg formation and development, embryonic mortality	[34,38]
RPL14B	Fertilization	[35]
RPS2	Egg formation and development	[34]
RPS7	Embryos developmental anomalies	[39]
RPL27a	Spermatogonia apoptosis	[40]
MRPL24	Developmental delays	[41]
RPS9M	Gametogenesis and seed development	[42]
RPS15A	Ovarian cancer	[43]
RPL39	Trophoblast cell proliferation	[44]
RPS14	Sertoli cell proliferation	[45]
RPL5	Breast cancer	[46]
RPLP0, RPLP1, RPLP2	Breast and ovarian cancer	[47]
RPS27L	Breast cancer	[48]

## Data Availability

This manuscript contains no experimental data, and all referenced literature is sourced from the PubMed database.

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
