# Peer review of "Research Progress of Ribosomal Proteins in Reproductive Development"

_ijms, 2024, doi:10.3390/ijms252313151_

Round 1

Reviewer 1 Report

Comments and Suggestions for Authors

Ribosomal proteins are key components of ribosomes, involved in regulating cellular processes such as nucleolar stress, oxidative stress, cell proliferation, and autophagy, which are crucial for oocyte and embryo development. This review highlights their role in reproductive development, the need to understand their mechanisms of action, and their potential as diagnostic markers and therapeutic targets for reproductive health-related diseases. Targeting ribosomal proteins, particularly those overexpressed in ovarian and endometrial cancers, could offer new biomarkers and therapeutic strategies for reproductive and developmental diseases linked to cellular dysfunctions.

Although I am not a direct specialist in the specific biochemistry and physiology of ribosomal proteins, I read the manuscript with interest. The review is well-written in terms of both language and scientific content.

Author Response

Ribosomal proteins are key components of ribosomes, involved in regulating cellular processes such as nucleolar stress, oxidative stress, cell proliferation, and autophagy, which are crucial for oocyte and embryo development. This review highlights their role in reproductive development, the need to understand their mechanisms of action, and their potential as diagnostic markers and therapeutic targets for reproductive health-related diseases. Targeting ribosomal proteins, particularly those overexpressed in ovarian and endometrial cancers, could offer new biomarkers and therapeutic strategies for reproductive and developmental diseases linked to cellular dysfunctions.

Although I am not a direct specialist in the specific biochemistry and physiology of ribosomal proteins, I read the manuscript with interest. The review is well-written in terms of both language and scientific content.

Agree. We agree with this comment. Thank you for taking the time to review our manuscript despite your busy schedule.Thank you for taking the time to review our manuscript despite your busy schedule.

Reviewer 2 Report

Comments and Suggestions for Authors

An exceptionally accessible and well-referenced overview of the contributions by ribosomal proteins to biological control that is operative in several key parameters of reproductive biology and pathology is presented. The topic is unquestionably important and there is a requirement for an overview of current understanding for engagement of ribosomal proteins in physiological regulation. The coverage of potential therapeutic targets is informative. The topic is covered from a balanced perspective and there is confidence that the review will be valued by scientists as well as clinicians.

Author Response

An exceptionally accessible and well-referenced overview of the contributions by ribosomal proteins to biological control that is operative in several key parameters of reproductive biology and pathology is presented. The topic is unquestionably important and there is a requirement for an overview of current understanding for engagement of ribosomal proteins in physiological regulation. The coverage of potential therapeutic targets is informative. The topic is covered from a balanced perspective and there is confidence that the review will be valued by scientists as well as clinicians.

Agree. We agree with this comment. Thank you for taking the time to review our manuscript despite your busy schedule. Thank you for taking the time to review our manuscript despite your busy schedule.

Reviewer 3 Report

Comments and Suggestions for Authors

The manuscript presents a comprehensive review of the role of ribosomal proteins in reproductive development, addressing their involvement in processes such as nucleolar stress, oxidative stress, cell proliferation, and autophagy. It also explores the potential diagnostic and therapeutic implications of these proteins. The review is well-structured, informative, and offers a valuable resource for researchers in the field.

The paper is clearly written and follows a logical structure, making it easy to follow the arguments presented. It references relevant studies, providing a detailed overview of current knowledge on ribosomal proteins in reproductive development. The inclusion of visual elements such as figures and tables enhances understanding and supports the text.

However, the paper would benefit from a dedicated Materials and Methods section. This would outline how the articles were selected, the databases used, and the specific inclusion and exclusion criteria. For example:

-              Which database was used?

-              Were there restrictions on publication dates or languages?

-              Were animal studies included alongside human studies?

Furthermore, the following typos emerged during the reading:

-              Paragraph 3.1.1: In the sentence “Knockdown of RPS27a reducing its interaction with RPL11 but promotes…arrest and apoptosis [53].” reducing should be replaced with reduces to ensure verb agreement within the sentence.

-              Paragraph 3.2: The sentence “Specifically, knockdown of RPL10 resulted in a reduction of TCF4/LEF1 protein levels, effectively inhibiting the WNT/β-catenin pathway. [76].” ends with two full stops. Remove the one before the citation.

In conclusion, the manuscript is a significant contribution to the field, providing a detailed and organized review of an important topic. With the inclusion of a Materials and Methods section and minor refinements, the paper will achieve even greater clarity and impact.

Author Response

The manuscript presents a comprehensive review of the role of ribosomal proteins in reproductive development, addressing their involvement in processes such as nucleolar stress, oxidative stress, cell proliferation, and autophagy. It also explores the potential diagnostic and therapeutic implications of these proteins. The review is well-structured, informative, and offers a valuable resource for researchers in the field.

The paper is clearly written and follows a logical structure, making it easy to follow the arguments presented. It references relevant studies, providing a detailed overview of current knowledge on ribosomal proteins in reproductive development. The inclusion of visual elements such as figures and tables enhances understanding and supports the text.

However, the paper would benefit from a dedicated Materials and Methods section. This would outline how the articles were selected, the databases used, and the specific inclusion and exclusion criteria. For example:

  Which database was used?

The references and data we used are drawn from the PubMed database.

Were there restrictions on publication dates or languages?

To deliver a comprehensive review of ribosome research in reproductive development, we collected and analyzed relevant literature available on PubMed. While we did not impose restrictions on the publication dates, all references included are in English.

Were animal studies included alongside human studies?

As stated in our review, we have summarized relevant studies on ribosomes in reproductive development in both animals and humans.

Furthermore, the following typos emerged during the reading:

Paragraph 3.1.1: In the sentence “Knockdown of RPS27a reducing its interaction with RPL11 but promotes…arrest and apoptosis [53].” reducing should be replaced with reduces to ensure verb agreement within the sentence.

Thank you for your valuable suggestion, we have made the requested revisions.

 Paragraph 3.2: The sentence “Specifically, knockdown of RPL10 resulted in a reduction of TCF4/LEF1 protein levels, effectively inhibiting the WNT/β-catenin pathway. [76].” ends with two full stops. Remove the one before the citation.

In conclusion, the manuscript is a significant contribution to the field, providing a detailed and organized review of an important topic. With the inclusion of a Materials and Methods section and minor refinements, the paper will achieve even greater clarity and impact.

Thank you for your valuable suggestion, we have made the requested revisions.